# The effect of marine ice-nucleating particles on mixed-phase clouds

Tomi Raatikainen[1], Marje Prank[1], Jaakko Ahola[1], Harri Kokkola[2], Juha Tonttila[2], and
Sami Romakkaniemi[2]

[1]Finnish Meteorological Institute, Helsinki, Finland
[2]Finnish Meteorological Institute, Kuopio, Finland

**Correspondence:** Tomi Raatikainen (tomi.raatikainen@fmi.fi)

**Abstract.** Shallow marine mixed-phase clouds are important for the Earth's radiative balance, but modelling their formation and dynamics is challenging. These clouds depend on boundary layer turbulence and cloud top radiative cooling, which is related to the cloud phase. The fraction of frozen droplets depends on the availability of suitable Ice-Nucleating Particles (INPs), which initiate droplet freezing. While mineral dust is the dominating INP type in most regions, high-latitude boundary layer clouds can be dependent on local marine INP emissions, which are often related to biogenic sources including phytoplankton. Here we use high resolution large eddy simulations to examine the potential effects of marine emissions on boundary layer INP concentrations and their effects on clouds. Surface emissions have a direct effect on INP concentration in a typical well-mixed boundary layer whereas a steep inversion can block the import of background INPs from the free troposphere. The importance of the marine source depends on the background INP concentration, so that marine INP emissions become more important with lower background INP concentrations. For the INP budget it is also important to account for INP recycling. Finally, with the high-resolution model we show how ice nucleation hotspots and high INPs concentrations are focused on updraught regions. Our results show that marine INP emissions contribute directly to the boundary layer INP budget and therefore have an influence on mixed-phase clouds.

## 1 Introduction

Stratocumulus clouds are shallow and thin clouds that cover large parts of the oceans and for this reason they have a significant effect on the radiative balance (Wood, 2012). Large uncertainties are related to mixed-phase clouds which contain both liquid cloud droplets and frozen particles (Korolev et al., 2017). Although this state is unstable as ice crystals tend to grow with the expense of liquid droplets, boundary layer mixed-phase clouds can persist for several hours or even days (Morrison et al., 2012). Ice crystal number concentration is important for the balance as too high concentration will lead to cloud glaciation (Murray et al., 2021).

Heterogeneous ice formation (or nucleation) means that a solid seed called Ice-Nucleating Particle (INP) is needed for the ice formation (Murray et al., 2012; Kanji et al., 2017). Ice crystals can form by deposition of water vapour on a dry particle or the freezing can start at the surface of an insoluble particle immersed in a liquid droplet (Hoose and Möhler, 2012). Contact nucleation refers to a case where freezing happens right after a collision between liquid droplet and an INP (Ladino Moreno

et al., 2013). Immersion freezing is the dominating primary ice nucleation mode for the temperature (253–263 K) and humidity conditions (saturation with respect to liquid water) in typical marine mixed-phase clouds (Murray et al., 2012).

Models based on the classical nucleation theory (e.g., Khvorostyanov and Curry, 2004; Chen et al., 2008; Hoose et al., 2010) can be adjusted to match with laboratory observations from typical freezing experiments (e.g., Murray et al., 2011; Hoose and Möhler, 2012). This adjustment relies mostly on INP specific parameters such as the contact angle (Chen et al., 2008; Ervens

and Feingold, 2012, 2013; Ickes et al., 2017). Using these parametrizations in simulating ice formation is complicated by the fact that the ambient INP population is a complex mixture of different chemical species. In practise, there is not enough observational information about the ambient INP population (INP types and their size distributions) so that the ice crystal concentration could be predicted just by using the Classical Nucleation Theory (CNT).

INP in a stochastic CNT-based model means a particle that carries the nucleating substrate, but INP in typical observations

is a particle that initiates droplet freezing at set conditions inside the instrument (e.g., Hartmann et al., 2020). Although direct comparison between these two INP definitions would require calculations, observations do provide some constrains for the models. Typical observations show that the logarithm of INP concentration decreases linearly with increasing temperature, but the absolute values vary by several orders of magnitude depending on the sampled air masses. For example, globally observed INP concentrations range from below detection limit up to $10\,\mathrm{L^{-1}}$ at 258 K temperature (Murray et al., 2021). The

highest values are seen over continents and the lowest in remote marine regions such as the Southern Ocean. Values close to $0.1\,\mathrm{L^{-1}}$ and below (at 258 K) are typical for marine boundary layers (McCluskey et al., 2018c; Wex et al., 2019; Gong et al., 2020; Hartmann et al., 2020, 2021), although episodic events related to either marine or terrestrial sources may increase concentrations by an order of magnitude (McCluskey et al., 2018c; Sanchez-Marroquin et al., 2020).

The most important INPs for shallow boundary layer clouds include dust and biogenic particles (Hoose et al., 2010). Desert

dust is globally the most common INP type, and it includes several different mineral compositions mainly related to their source regions (Boose et al., 2016; Kok et al., 2021). There are also dust sources specific for the cold high-latitude environments (Bullard et al., 2016) and they could be important local INP sources in the absence of desert dust from the mid or low latitudes (Tobo et al., 2019; Sanchez-Marroquin et al., 2020). Relatively high dust concentrations can be seen in continental outflow regions, but concentrations are significantly lower in the remote marine regions such as the Southern Ocean (Prospero et al.,

2002; Kok et al., 2021). Locally emitted biogenic marine INPs can be important especially for these remote regions (e.g., Burrows et al., 2013; Wilson et al., 2015; DeMott et al., 2016; Vergara-Temprado et al., 2017; Huang et al., 2018; McCluskey et al., 2018c; Hartmann et al., 2020, 2021; Huang et al., 2021).

The current view is that marine INPs are emitted as primary particles as a part of sea spray aerosol (SSA). At moderate wind speeds, SSA is produced mainly by bubbles bursting at the sea surface (Mårtensson et al., 2003). Soluble sea salt aerosol is

generally quite poor INP, but sea spray contains other material from the sea surface layer that may initiate droplet freezing. Although, dust may be re-emitted from the sea surface (Cornwell et al., 2020), current focus is on biogenic or organic material. There are experiments showing that artificially generated SSA contains INPs (Wilson et al., 2015; DeMott et al., 2016; McCluskey et al., 2018b; Wolf et al., 2020; Gong et al., 2020; Ickes et al., 2020; Mitts et al., 2021) and studies on ambient INPs linked to marine origin (McCluskey et al., 2018a, c; Hartmann et al., 2020, 2021). Most of these studies link INPs to

phytoplankton biological activity, which is typically related to chlorophyll concentrations. The actual INPs can be composed of molecules, intact cells, or microbe fragments (Burrows et al., 2013; McCluskey et al., 2018b; Knopf et al., 2018).

The details of the marine INPs related to their origin, emission rates and ice nucleation properties are still highly unclear. Nevertheless, there are a few large-scale studies exploring the potential importance of marine INPs on shallow clouds (e.g., Burrows et al., 2013; Wilson et al., 2015; Vergara-Temprado et al., 2017; Huang et al., 2018; McCluskey et al., 2019; Zhao et al., 2021). Global simulations by Burrows et al. (2013) showed that marine emissions could lead to mean INP concentrations up to $0.01$–$0.02\,\mathrm{L^{-1}}$ (at 258.15 K) in the high-latitudes. For the southern high-latitudes this means that marine and dust INPs have roughly equal contributions while dust sources dominate concentrations by about a factor of ten in the northern high-latitudes. Global simulations by Wilson et al. (2015) and Vergara-Temprado et al. (2017) showed similar spatial trends and concentrations for the marine INPs. Huang et al. (2018) showed the importance of marine INPs on droplet freezing at the mixed-phase cloud temperature range, where their contribution may exceed 50 %. However, they also note that the result depends much on the model assumptions. McCluskey et al. (2019) used their previously developed marine INP parametrization in a global simulation and compared their predictions with observations from the North Atlantic and the Southern Ocean. They concluded that SSA is often the dominant INP source in both remote marine environments. Zhao et al. (2021) showed that marine INPs dominate primary ice nucleation over the Southern Ocean and Arctic boundary layer, while dust INPs are more abundant elsewhere. In general, these studies support the view on important role of marine INPs in remote high-latitude regions and at low altitudes. These regions are dominated by low-level mixed-phase clouds, which are known to be problematic for large scale models due their coarse resolution. Cloud resolving models are more suitable tools for exploring the effects of boundary layer dynamics on marine and dust INPs, and their interactions with clouds.

In this study we explore the potential effects of marine INPs on mixed-phase boundary layer clouds by using UCLALES-SALSA, which is a cloud resolving large eddy simulator (LES) coupled with detailed aerosol-cloud-ice microphysics (Tonttila et al., 2017; Ahola et al., 2020). Specifically, we will examine how marine INP emissions impact boundary layer INP concentrations and vertical distributions when compared with the effects of background dust aerosol and dust entrained from the free troposphere. We will also examine the impacts of INPs on mixed-phase cloud dynamics and stability. Our results can be used to prioritize additional processes to be included in large scale models.

## 2    Methods

Previously, Ahola et al. (2020) used the LES model inter-comparison study described by Ovchinnikov et al. (2014) for testing and validating the newly implemented UCLALES-SALSA ice microphysics. Here we use the same model and simulation settings as described in detail by Ahola et al. (2020) except with a few modifications explained below. First, we will briefly describe the original case study by Ovchinnikov et al. (2014), which is based on observations from the Indirect and Semi-Direct Aerosol Campaign (ISDAC) focused on Arctic mixed-phase clouds. Then we describe the current LES model with a focus on ice microphysics.

## 2.1 The ISDAC case study

ISDAC observations are described in detail by McFarquhar et al. (2011). Briefly, the campaign took place in the vicinity of Utqiaġvik (formerly known as Barrow) located at the North coast of Alaska (USA) near the Arctic Ocean during April 2008. The focus was on aircraft observations, but ground observations such as balloon-borne soundings were conducted at a research site near Utqiaġvik. The research aircraft was equipped with various aerosol and cloud instruments for measuring size distributions and chemical composition. In-cloud Ice Crystal Number Concentrations (ICNCs) and ice crystal shapes were also measured.

Different cloud types and conditions were seen during ISDAC, of which Ovchinnikov et al. (2014) used observations from 26 April 2008 to derive their semi-idealized LES setups. The initial state was described by vertical temperature, humidity and wind profiles, and a bimodal ammonium bisulfate aerosol size distribution. The run-time settings include simplified microphysics (disabled all collision processes, no warm rain, and assuming spherical low-density ice particles), parametrized radiation scheme, large scale subsidence based on a constant divergence ($Q = 1.5 \cdot 10^{-6}\,\mathrm{s}^{-1}$), zero surface sensible and latent heat fluxes, and a weak nudging of winds and free tropospheric humidity and temperature towards their initial values. During this specific day, the research aircraft sampled single-layer mixed-phase stratiform cloud, which persisted for 15 hours over the ice-covered Arctic Ocean. Most ice particles were pristine dendrite crystals and drizzle was absent, which justified the exclusion of ice aggregation and warm rain processes. Model domains cover $3.2\,\mathrm{km}$ (64 grid cells with $50\,\mathrm{m}$ resolution) in both horizontal dimensions and $1.5\,\mathrm{km}$ (140 grid cells with $10\,\mathrm{m}$ resolution below $1200\,\mathrm{m}$ and stretched grid after that) in the vertical dimension. Ovchinnikov et al. (2014) used diagnostic ice nucleation scheme, where the in-cloud ICNC was tuned to match with a pre-defined value. They set the baseline ICNC to $1\,\mathrm{L}^{-1}$ based on the observations. This value is slightly larger than the concentration range of 0.15–0.66 $\mathrm{L}^{-1}$ reported by Hiranuma et al. (2013), but well within the typical variability.

Ahola et al. (2020) used these settings with the exception that they used the newly implemented prognostic ice nucleation scheme to predict the ICNC in their UCLALES-SALSA simulations. The freezing rates were predicted using a stochastic CNT-based immersion freezing parametrization. The other ice nucleation modes are not relevant for the simulated cloud conditions, so they were ignored. The observed cloud state and temperatures below 263 K do not favour secondary ice production, so it too was ignored. Because ISDAC observations provided little information about the INP size distribution, chemical composition or ice nucleation efficiency, the practical approach for the simulations was aiming to produce similar ICNCs as in Ovchinnikov et al. (2014) by adjusting the total INP concentration while keeping the ice nucleation parameters fixed. In practise, an adjustable fraction of the initial aerosol was considered as dust-containing background INPs. This is a computationally efficient approach when the focus is on aerosol-cloud-ice interactions instead of the details of the freezing mechanism. We will also use this approach with slight modifications and an additional marine INP source. More details about this are given in the next section.

We made one modification to the initial temperature and humidity profiles used in the previous case studies. The initial temperature and humidity profiles represented de-coupled marine boundary layer, but eventually the boundary layer became coupled in the model simulations (Ovchinnikov et al., 2014). To allow vertical mixing from the beginning, we initialize our

simulations with well-mixed profiles (constant liquid water potential temperature and total water mixing ratio in the boundary layer). The effect of boundary layer de-coupling will be examined in Sect. 3.5.

## 2.2 LES modelling

Current simulations are made with large eddy simulator UCLALES-SALSA. This model is based on the commonly used UCLALES (Stevens et al., 1999, 2005; Stevens and Seifert, 2008) where cloud microphysics is replaced by the SALSA aerosol module (Kokkola et al., 2008, 2018) extended for warm (Tonttila et al., 2017) and mixed-phase (Ahola et al., 2020) clouds. Because UCLALES-SALSA has been described in previous publications, only a brief description focusing on SALSA and the current ice microphysics scheme is given here.

Aerosol, cloud droplet and ice particle chemical composition (here just water, dust and sulphate) and size distributions are described using sectional approach based on dry particle size bins. Water is the substance that partitions between vapour and condensed phases, dust is the insoluble ice-nucleating material, and sulphate is a soluble substance. The model has just one species (dust) with ice nucleation ability, so it is used to describe both background mineral dust and marine biogenic emissions. Although the species is called dust, it can be used to represent any ice-nucleating material by adjusting its ice nucleation parameters such as the contact angle. Details of the ice nucleation scheme are given later in this section. Water and dust physical properties had their default values, but sulphate density, molecular weight and dissociation factor were set to $1780\,\mathrm{kg\,m^{-3}}$, $115.11\,\mathrm{g\,mol^{-1}}$ and 2.0, respectively, so that it represents ammonium bisulfate.

Water vapour partitioning is based on diffusion limited non-equilibrium droplet or ice crystal growth, except that the equilibrium water content is assumed for aerosol when RH<98 %. For the supersaturated regions, the non-equilibrium droplet growth determines cloud activation, which takes place when aerosol wet size exceeds the critical droplet size. Cloud activation means that the activated aerosol is moved from the aerosol bin to the corresponding cloud bin with matching dry particle size. Cloud (or aerosol) droplet freezing is modelled based on a stochastic (time-dependent) immersion freezing parametrization described below. By default, the freezing is limited to cloud droplets, but the effect of allowing aerosol freezing (interstitial and those outside clouds) will be examined in Sect. 3.5. The immersion freezing parametrization predicts the number of frozen cloud (or aerosol) droplets in each size bin during the model time step, and the newly formed ice crystals are moved from the cloud (or aerosol) bins to the corresponding ice bins. From the common cloud microphysical processes, only the ice crystal sedimentation was enabled in the default simulations. However, we will test the effect of allowing aerosol and cloud droplet sedimentation in Sect. 3.5.

The initial bimodal ammonium bisulfate (called sulphate in SALSA) aerosol size distribution from Ovchinnikov et al. (2014) and the dry size bin limits are shown in Fig. 1. For the current simulations, the first SALSA particle size range for the nucleation mode (bins 1–3) covers dry diameters from 3 to 20 nm while the default upper limit is 50 nm. The second size range (bins 4–15) covers dry diameters from 20 nm to 10 μm using twelve bins instead of the default of seven. This improves the size resolution for cloud droplets and ice particles, which are present only on the second size range. Externally mixed dust-containing INPs are described by using another set of bins from the second size range for aerosol, cloud droplets and ice particles (so-called b-bins). In practice, the a-bins describe the distributions of sulphate aerosol and related cloud droplets (no ice in the a-bins due to the

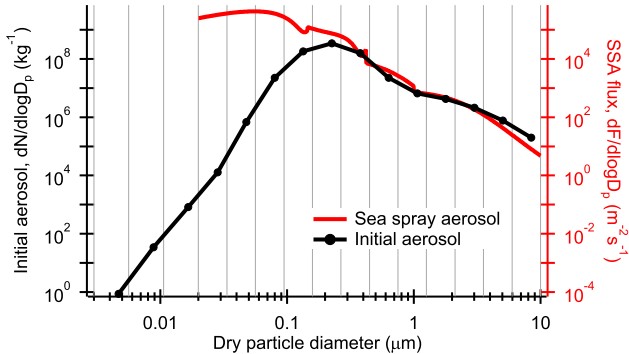

**Figure 1.** Initial aerosol size distribution for the ISDAC case (black line, left axis) and parametrized sea spray aerosol (SSA) emission flux for $6\,\mathrm{m\,s^{-1}}$ wind speed and $271.15\,\mathrm{K}$ temperature (red line, right axis). Vertical grey lines represent aerosol dry size bin limits for SALSA

absence of ice nuclei), and b-bins describe dust–sulphate aerosol and related cloud droplets and ice crystals. It should be noted that all dust-containing particles in these simulations are called INPs, because they have a non-zero freezing probability based on the stochastic ice nucleation scheme. However, experimental studies often report INPs as those particles that initiate ice formation under specific instrument conditions (typically temperature, humidity and residence time). Unless otherwise stated, we follow the dust-related definition here.

Here we use an upgraded version of UCLALES-SALSA where size-dependent sea spray aerosol (SSA) emissions are parametrized as a function of the domain mean wind speed at the height of 10 m and sea surface temperature (here constant 271.15 K). The parametrization is valid for open ocean, so we ignore the effects of sea ice and nearby coasts in these simulations. For the dry particle size range 0.020–1 μm the SSA emission parametrization is from Mårtensson et al. (2003) and for the 1–10 μm size range it is from Monahan et al. (1986). For the latter size range, the temperature dependency term is from

Jaeglé et al. (2011). Emissions of particles smaller than 20 nm are ignored as these have a negligible role for clouds. Figure 1 shows the SSA flux for $6\,\mathrm{m\,s^{-1}}$ wind speed and 271.15 K sea surface temperature as an example. The initial background aerosol size distribution is based on observations by Earle et al. (2011) covering a size range from 100 nm to about 10 μm, which explains the relatively low concentrations of sub-100 nm particles as the the aerosol size distribution is fitted based on larger particles. These sub-100 nm particles have a minor role for clouds, because there are enough larger particles.

Different ice nucleation modes were implemented as described by Ahola et al. (2020), but here the focus is on immersion freezing (Appendix A in Ahola et al. (2020)). It is based on the classical theory of heterogeneous ice nucleation where the ice formation takes place at the surface of a solid insoluble substrate immersed in super-cooled liquid droplet as presented by Khvorostyanov and Curry (2000). Droplet freezing in this classical approach is a stochastic process, so the parametrization predicts the droplet freezing rate which is integrated over time to obtain the number of newly formed ice crystals. Ice nucle-

ation rate depends mainly on ambient conditions (temperature and relative humidity) and the properties of the solid insoluble substrate (size and compound-specific ice nucleation parameters). In our simulations, dust is the solid insoluble substrate and it is mixed with soluble sulphate (equal dry particle volume fractions) to allow the aerosol and cloud droplet water uptake. All

other ice nucleation parameters in the immersion freezing parametrization had their default values, except cosine of the contact angle was increased from 0.50 to 0.57 to enhance freezing at these relatively high temperatures (see Appendix A in Ahola et al. (2020)). This value is within the range of 0.36–0.73 representing surface soil, quartz and sand (Khvorostyanov and Curry, 2000). Assuming a constant contact angle means that the ice nucleation parametrization is valid for a narrow temperature range which in our case means in-cloud temperatures of about 258 K. For this reason, we are not examining temperature dependency, but focus on the 258 K temperature.

Since we are limited to two externally mixed particle populations (one for the soluble sulphate particles and one for the INPs), marine and background INPs need to be described with the same INP population. This means that they will have the same chemical composition (dust–sulphate) and ice nucleation parameters (contact angle). Because we cannot adjust ice nucleation parameters independently, we will keep those fixed and adjust the emissions and initial concentrations of the marine and background INPs, respectively. Although their size distributions could be adjusted, we will use the background and sea-spray aerosol size distributions (Fig. 1) in the absence of suitable observations. This leaves us two adjustable parameters: the fraction of INPs in the initial background aerosol and the fraction of INPs in the wind speed dependent sea-spray aerosol flux. Due to this model limitation, we also assume that the non-INP sea spray aerosol (SSA) is composed of sulphate so that the emissions will influence the soluble background particles without changing their chemical composition. This assumption has a minor effect on clouds, because there are enough efficient cloud condensation nuclei.

Our strategy for adjusting the fractions of INPs in the background aerosol and SSA flux is based on the fact that some amount of INPs are needed to maintain the mixed-phase cloud and that marine INP emissions may have a dominant role in certain conditions (not necessarily ISDAC). In the following simulations we will examine the potential effects of marine INPs on mixed-phase clouds with wide range of background INP concentrations. Our previous simulations show that ICNC in the order of $4\,L^{-1}$ may cause complete cloud glaciation (Ahola et al., 2020), so this is suitable upper limit for our simulations. This concentration is about an order of magnitude larger than seen in observations by Hiranuma et al. (2013). Preliminary test simulations showed that this cloud glaciation limit is approached when the initial background aerosol INP number fraction is set to 0.00010. Setting the sea spray INP fraction to 0.005 showed that this is large enough to produce a notable effect even without the background INPs while not exceeding the glaciation limit with high concentrations of background INPs. Therefore, the upper limit for initial background aerosol INP number fraction is set to 0.00010, and values smaller than this (down to zero) will be used in other simulations. The effects of marine INP emissions are examined by setting the INP emissions on (fraction 0.005) and off (fraction 0.0). With this approach we will have a range of simulations representing INP sources from purely marine to purely background, and mixed sources with different background INP concentrations.

All simulations have the same initial background aerosol size distribution (Fig. 1) and the wind speed dependent (constant sea surface temperature) SSA flux is switched on after the one-hour spin-up. These influence the soluble sulfate aerosol population (a-bins). Background and marine INPs are represented by additional initial aerosol INP population and sea surface INP flux, respectively, influencing the dust–sulfate aerosol (INP) population (b-bins). With this approach, the only difference between simulations is the INP concentration.

Although the ice nucleation scheme described above considers the INP concentration as an adjustable parameter, we can examine if at least the maximum initial background INP concentration is reasonable based on observations from ISDAC and elsewhere. McFarquhar et al. (2011) reported highly variable clear-air INP concentrations ranging from less than 1 to more than $10\,L^{-1}$ measured with a Continuous Flow Diffusion Chamber (CFDC), but instrument conditions were not specified (RH between 100 and 110 % and temperatures from 243 to 263 K). In order to compare measured and our INP concentrations, we need to calculate the ice crystal concentration that would result in an exposure of our aerosol size distributions to certain temperature and humidity. For this we assume a typical CFDC instrument residence time of 10 s and use our cloud top temperature (258 K) and relative humidity (100 %), and assume $1\,kg\,m^{-3}$ air density for the unit conversions. For the maximum initial background INP aerosol size distribution, the calculated ice crystal concentration would be $1.8\,L^{-1}$, which is at the high end of the observational range and representative of continentally-influenced air masses. This suits well with our purpose of examining the full range of background INP concentrations by conducting a series of simulations with lower concentrations.

Comparing our marine INP fluxes with observations or other simulations representing continuous emissions is limited by the fact that the fluxes in our simulations start after the one-hour spin-up and then have limited time for an impact. Therefore, our fluxes must be significantly higher than any continuous emissions. Based on the simulated impacts on clouds (shown in the next section), the current INP emissions seem to be reasonably high as they alone can maintain the mixed-phase cloud. On the other hand, the emissions are low enough so that their contribution is small in the presence of high concentrations of background INPs.

## 3 Results and discussion

### 3.1 Cloud response to INPs

As the first step we made eight simulations where marine INP emissions were either on or off and the initial background aerosol INP number concentration had four different values. Marine INP emissions are specified as a fraction of the SSA flux, and here emissions on and off mean fractions 0.005 and 0.0, respectively. Initial background INP number concentrations are specified as a fraction of the initial aerosol, and here the cases are called zero (fraction is 0.0), low (0.00001), medium (0.00005), and high (0.00010). Background INP fractions were selected so that the simulations without marine INP emissions cover the range from an ice-free case up to a cloud that is becoming mostly glaciated. When marine INP emissions are switched on, the fraction of INPs in the sea spray aerosol is high enough to have an impact on clouds. Simulation time was set to 24 h including a 1 h spin-up for SSA emissions and a 2 h spin-up for ice microphysics. Because most adjustments take place during the first 10 h and the trends are steady after 12 h, we will focus on the first 16 hours.

Results from our simulations are shown in Fig. 2. Cloud base and top heights are the domain minimum base and maximum top heights, respectively, so they represent the full extent of the cloud deck. Ice crystal number concentrations are averaged over grid cells where ice mass mixing ratio exceeds $1 \cdot 10^{-8}\,kg\,kg^{-1}$. Liquid (LWP) and ice (IWP) water paths are domain mean values. Simulations where marine INP emissions are switched on and off are shown with the solid and dashed lines, respectively. The effect of marine INP emissions is clearly seen in the initial ice crystal number concentration trends, which

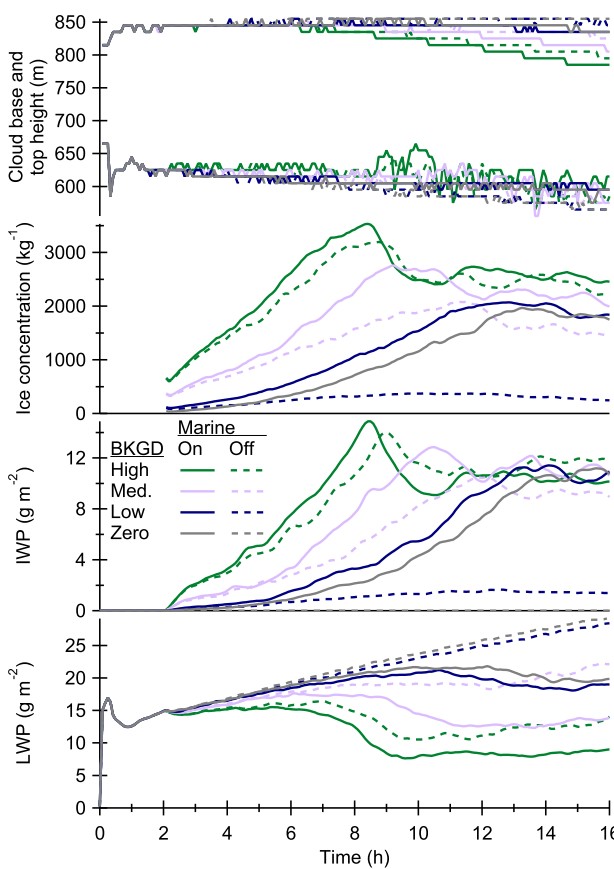

**Figure 2.** Time series of cloud base and top heights, ice crystal number concentration, and ice (IWP) and liquid (LWP) water paths from the eight model simulations with different background (BKGD) aerosol INP concentrations (zero, low, medium, and high) and marine INP emissions switched on (solid lines) or off (dashed lines).

means that the aerosol is effectively transported from the sea surface up to the cloud layer (between $600\,\text{m}$ and $850\,\text{m}$). IWP depends mostly on ice crystal number concentration because the mean ice crystal diameters are similar in all simulations ($400$–$430\,\mu\text{m}$ at 10 h). Condensible water is limited so an increase in IWP is seen as a decrease in LWP. For this reason, ice number concentration is the most important parameter for these mixed-phase clouds.

The largest differences between simulations with marine INP emissions on or off are seen with the lowest background INP concentrations. This means that marine INP emissions become more important with decreasing background INP concentration. In addition, marine INP emissions alone seem to produce the same final mixed-phase cloud state as having the medium or high INP background without marine INP emissions. There are two simulations, both without marine INP emissions, where the INP concentration is so low that the result is a thick almost purely liquid cloud. The other simulations end up to ice number concentration of about $2000\,\text{kg}^{-1}$ (IWP about $10\,\text{g m}^{-2}$), because precipitation increases with increasing INP concentration.

This is the case even with the highest INP concentrations, but then there is also a reduction in LWP after the first 8 hours, which leads to a mostly glaciated cloud state.

The cloud starts to glaciate (LWP decreases and IWP increases) rapidly when ice crystal number concentration approaches $3000\,\text{kg}^{-1}$, which was already confirmed in our previous study (Ahola et al., 2020), but in this case the limit can be exceeded due to an additional INP sea surface source. The drop in the ice number concentration in these simulations after 8 h is related

to precipitation. In fact, the removal of INPs with precipitation saves the cloud from complete glaciation, but it also removes part of the condensible water. This has an impact on cloud stability as cloud top radiative cooling requires liquid water. The reduction in liquid water content explains why cloud top heights decrease in the three simulations with the lowest LWPs. Also, the decrease in LWP reduces ice formation preventing the cloud from complete glaciation even with additional INPs from the surface.

## 3.2 INP budget

Cloud development in these simulations depends mainly on ice crystal number concentration, which is related to the availability of INPs, so here we focus on the INP budget. Figure 3 shows the changes in column total INP mass (left) and number (right) concentrations due to the common removal and production mechanisms. The INP mass includes the total mass of the insoluble ice-nucleating material (represented by dust in the model simulations) in aerosol, cloud droplets and ice crystals. Calculations

cover the whole domain, but changes are negligible in the free troposphere. INP mass is related to large particles which are effective INPs, but number concentration depends mostly on small particles that are simply too small to be effective ice nuclei (size distribution shown in Fig. 1). This is not an issue for precipitation, because it includes only ice crystals. To have a more realistic estimate of the INP number budget, subsidence and surface fluxes include particles larger than 159 nm in dry diameter. This is the lower limit of the first size bin (Fig. 1) that has a significant fraction of ice in all our simulations. Specifically, at

least 90 % of the ice crystals originate from INPs larger than 159 nm during the first eight simulation hours. Because this limit is somewhat subjective and time and case dependent, we will focus more on the INP mass.

Precipitation is the main INP removal mechanism, and it can easily exceed production. In fact, the total INP mass is decreasing in all other simulations except the one without background INPs (Fig. 4). Surface INP emissions are a fraction (0.005) of the total SSA flux (Fig. 1), which depends on the 10 m wind speed (approx. 5–6 m s$^{-1}$) and sea surface temperature (fixed to

285 271.15 K). Changes in the 10 m wind speed cause the slow decrease in surface fluxes. Subsidence is described by a downward vertical velocity related to altitude ($z$) and the fixed large scale divergence $Q = 1.5 \cdot 10^{-6}\,\text{s}^{-1}$. Subsidence velocity ($= Qz$) is applied to all prognostic variables and it has an effect whenever there are vertical concentration gradients. Subsidence has a fairly small impact on INPs due to competing effects: while subsidence brings INP-rich aerosol from the free troposphere, it depletes boundary layer cloud and ice species at the same time (profiles discussed below). In addition, subsidence has a

290 negative effect when surface concentration is increased due to marine INP emissions. This is the reason for the clear decrease in INP number and also for the initial decrease in INP mass. Subsidence becomes significant INP source when the initially high boundary layer concentration decreases due to precipitation while that at the free troposphere stays high. However, subsidence

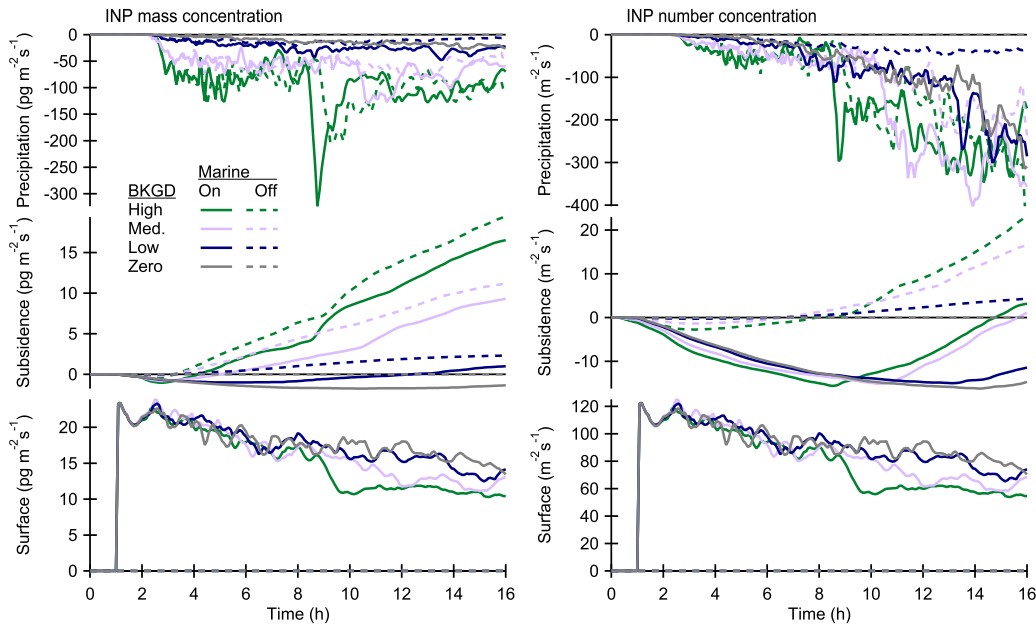

**Figure 3.** The three mechanisms (precipitation, subsidence and surface emissions) affecting on INP mass (left) and number (right) concentrations. Simulations are initialized with different background INP concentrations and marine INP emissions are switched on (solid lines) or off (dashed lines). The effects of surface emissions and subsidence on INP number concentration are calculated for particles larger than 159 nm in dry diameter.

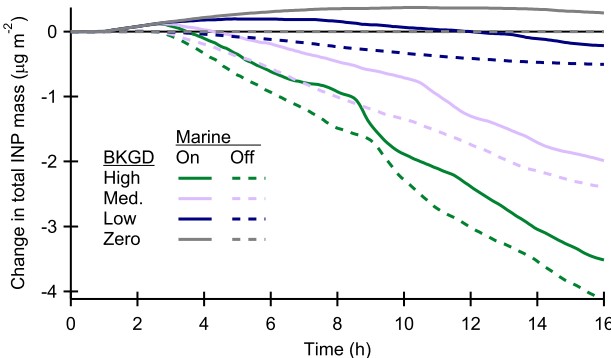

**Figure 4.** Changes in the column integrated INP mass from the initial values. The initial values for the four different background INP concentrations (high, medium, low and zero) are 9.6, 4.8, 1.0 and 0.0 $\mu g\,m^{-2}$.

continues to have a small influence when the initial INP concentration is low enough to avoid the rapid removal of boundary layer INPs.

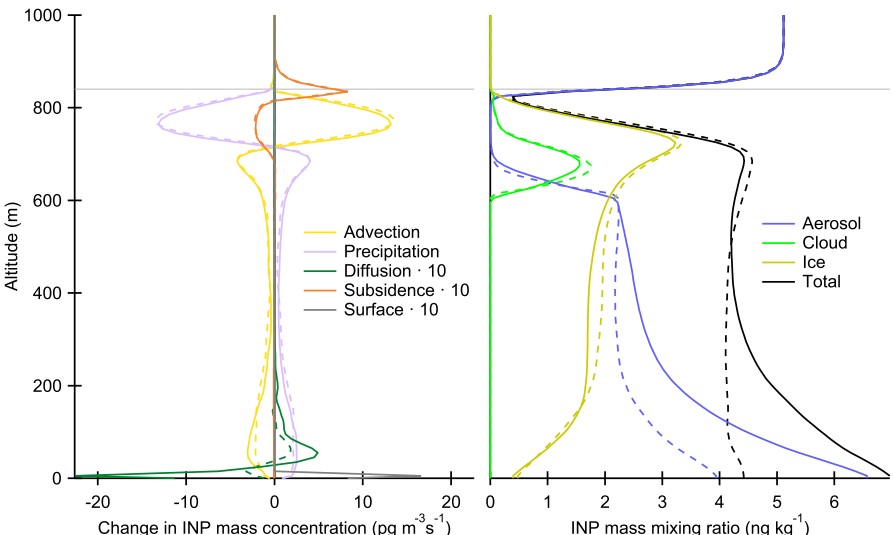

**Figure 5.** Horizontally averaged profiles of the main processes affecting on INP mass concentrations from simulations with (solid lines) and without (dashed lines) marine INP emissions (high background INP concentration). Diffusion, subsidence, and surface flux contribution are multiplied by a factor of ten for clarity. Corresponding INP mass concentrations in aerosol, cloud, ice and in total are shown in the right panel. The profiles are one-hour averages over the eighth simulation hour. Cloud top height (840 m) is indicated by the horizontal lines.

Figure 5 shows horizontally averaged profiles of the main processes affecting on vertical INP mass distributions from simulations with (solid lines) and without (dashed lines) marine INP emissions (high background INP concentration). Corresponding INP mass mixing ratios in each phase and in total are shown in the right panel. The profiles are one-hour averages over the eighth simulation hour of the instantaneous model outputs (produced after every 300 s; original units used). The purpose of the averaging is to reduce noise and fluctuations related to the instantaneous model outputs. The time interval was selected as an example, because the highest ice number concentrations are seen at that time just before precipitation rates increase significantly. The other time periods and simulations (different background INP concentrations and with and without marine INP emissions) show similar behaviour, but magnitudes of these processes depend mostly on ice crystal number concentration, which influences precipitation and eventually vertical INP distributions (see the description below).

Figure 5 emphasizes the important role of the vertical fluxes in recycling INPs compared with the relatively small contributions from sources and removal process (Fan et al., 2009; Solomon et al., 2015). Advection (aerosol, cloud, and ice) and precipitation (ice only) have the largest and almost opposite effects on the vertical INP distribution. Advection means mixing within the domain (i.e., no net effect on mass or number), so it practically reduces concentration differences caused by precipitation. Precipitation carries INPs from the cloud droplet freezing region to the near surface sublimation region where most INPs are released back to aerosol (more details in the next section). Advection and precipitation maintain steady profiles by recycling INPs while a fraction of particles is removed by surface precipitation and some particles are entering from the sea surface and free troposphere (subsidence).

Subsidence introduces aerosol particles from the free troposphere (positive values at the cloud top), but at the same time depletes cloud and ice species in the cloud (negative values below the cloud top). This is related to the steep gradient in total INP mass concentration (Fig. 5, right panel). When the total mass concentration is larger in the free troposphere than in the boundary layer (e.g., when precipitation removes the largest particles), the net effect of subsidence is positive (see Fig. 3 above). However, surface emissions change the concentration gradient so that subsidence has a negative effect near surface. In this case the difference between boundary layer and free troposphere dominates. In these simulations, subsidence and entrainment mixing are balanced so that the cloud top height is almost constant. Otherwise changes in the mixing layer depth would influence INP concentrations.

The only clear difference between simulations with and without marine INP emissions is seen in the near surface diffusion and surface emissions rates. Marine INP emissions influence only the first model layer, and sub-grid scale diffusion is the main mechanism transporting particles from the first model layer to the layers above where advection dominates. Diffusion reduces concentration differences within the domain just like advection, but diffusion is significantly weaker and limited to the lowest model layers due to the dependency on eddy diffusivity. Diffusion is not causing INP removal to the sea surface, because aerosol sedimentation (includes the effect of particle diffusivity) is disabled in these simulations. A test will be conducted in Sect. 3.5 where aerosol sedimentation is enabled.

The effect of marine INP emissions can be seen in the total INP mass concentration profiles as an increase near the sea surface. Interestingly, aerosol phase INP mass concentration profiles are similar with or without marine INP emissions. They both show decreasing trend with height above sea surface, which is typically related to a surface source. In this case, however, ice crystal sedimentation and sublimation near the surface is an additional reason. We will focus on this topic in the next section.

## 3.3 Ice budget

Because ice crystal mass and number concentrations are important for the time evolution of the cloud and the process are related to the INP budget, we will briefly examine the ice budget. Figure 6 shows the effects of the main production and removal mechanisms for ice mass mixing ratio (left) and ice crystal number concentration (right). Nucleation (freezing of cloud droplets) is the only mechanism producing new ice particles, but nucleation has a negligible contribution to the ice mass. Ice mass depends mainly on water vapour sublimation and deposition rates, but the importance of precipitation increases with time. Only the largest ice crystals survive the fall through the sublimation layer, which means that they are permanently removed by precipitation. Other ice crystals are moved back to aerosol bins when essentially all ice has been sublimated. Here subsidence reduces ice concentrations at the top of the ice layer by bringing ice-free air from above.

Figure 7 shows the key processes affecting on vertical ice mass (left) and number (middle) concentration profiles for the simulation with high background INP concentration and marine INP emissions switched on. The profiles are one-hour averages over the eighth simulation hour. The right panel shows normalized (by the maximum value) profiles of cloud water and ice mass and ice crystal number concentrations. Advection, precipitation and partitioning of water vapour (sublimation/deposition) have the largest effects on vertical ice mass distribution. Ice crystals grow by deposition of water vapour both in-cloud and

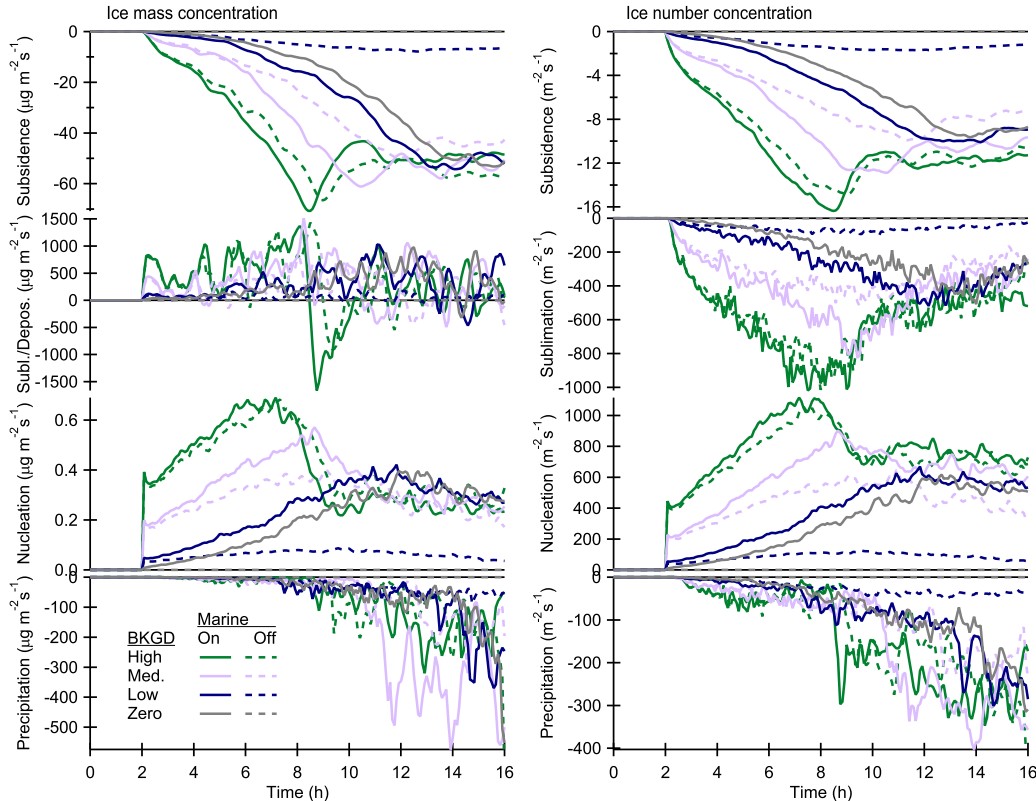

**Figure 6.** The four mechanisms (subsidence, water vapour sublimation and deposition, nucleation, and precipitation) affecting ice mass (left) and number (right) concentrations. Simulations are initialized with different background INP concentrations and marine INP emissions are switched on (solid lines) or off (dashed lines).

below cloud when RH with respect to ice exceeds 100 % (above 470 m) and sublimation takes place otherwise. Precipitation redistributes ice (and INPs) from the freezing and growth regions to the regions below. Advection reduces concentration differences caused by the other processes. Ice nucleation is important for the number concentration, and it takes place at the top of cloud where the lowest temperatures are seen. The smallest ice crystals may lose all ice in the sublimation region and in that case they are released back to aerosol (sublimation). This is an important INP source for the near-surface layer (mostly below 200 m), and most of the time sublimation rates exceed particle losses with precipitation (Fig. 6). Subsidence has the largest effect at the top of ice layer due to the steep gradient in mass and number concentrations. Diffusion has the largest effect on transporting ice from above to the lowest level where particle diffusivity increases precipitation removal rates.

Ice nucleation in these simulations is focused on the cloudy region and especially closer to the cloud top. However, ice nucleation is limited to cloud droplets, so aerosol freezing below and above the cloud and the freezing of interstitial aerosol are prohibited. Just like cloud droplets, a fraction of the aerosol contains an INP immersed in supercooled liquid, so the same

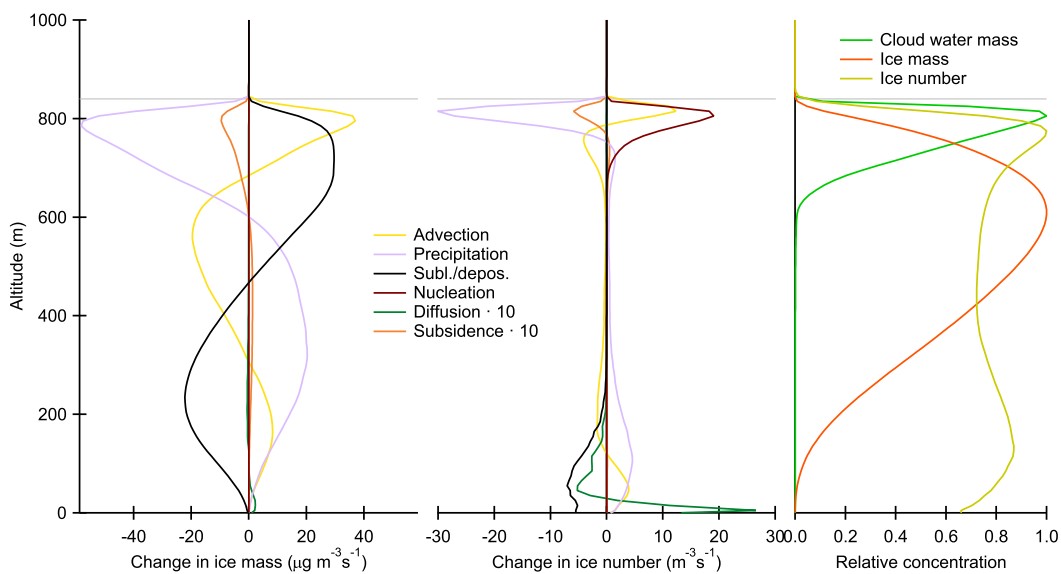

**Figure 7.** Horizontally averaged profiles of the main processes affecting on ice mass (left) and number (middle) concentrations. Diffusion and subsidence are multiplied by a factor of ten for clarity. Cloud water and ice mass mixing ratios and ice number concentration profiles normalized by their maximum values are shown in the right panel. The profiles are one-hour averages of the eighth simulation hour from the simulation with high background INP concentration and marine INP emissions switched on. Cloud top height (840 m) is indicated by the horizontal lines.

immersion freezing parametrization can be used to predict their freezing rates. The effect of aerosol freezing is tested in Sect. 3.5.

### 3.4 Details about cloud ice formation

In the above it was shown that advection transports INPs to the cloud top where most of the droplet freezing takes place. However, a closer look at 3D data shows that there is significant horizontal variability in the freezing rates and INP mass mixing ratios, and this variability can be best explained by vertical velocity. This is not a surprise knowing the importance of vertical velocity for cloud activation.

Figure 8 shows vertically integrated ice nucleation rates (the number of primary freezing events per second in a column of 365 air above $1 \mathrm{m}^2$ of sea surface area) as a function of mean in-cloud vertical velocity for each column of the domain from a single time step at 8 h (4096 columns in total). Marker colour is based on the corresponding vertically integrated column INP mass mixing ratio (the total INP mass in aerosol, cloud droplets and ice crystals above $1 \mathrm{m}^2$ of sea surface area). Figures on the left and right show simulations with and without marine INP emissions, respectively. Once again, marine INP emissions do not have a clear direct effect on nucleation rate but contribute indirectly via INP number concentration. This is the main reason for 370 the differences between these two figures.

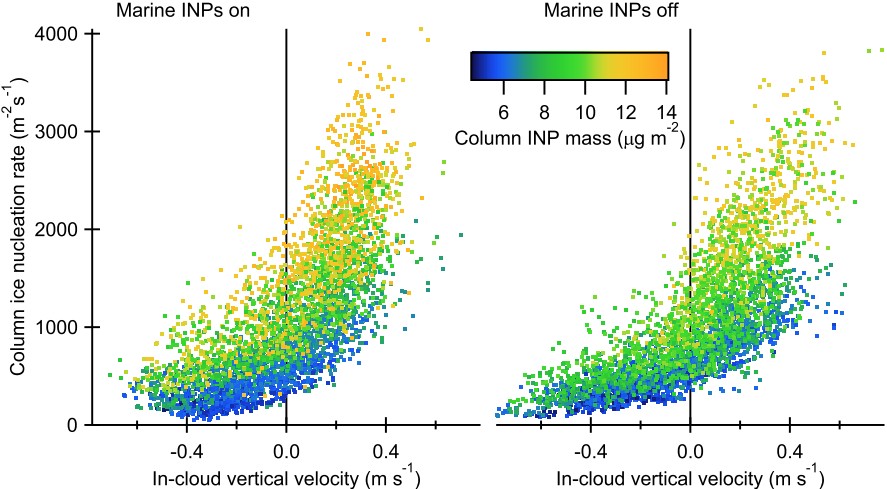

**Figure 8.** Vertically integrated ice nucleation rate (the number of primary freezing events per second in a column of air above $1\,\mathrm{m}^2$ of sea surface area) as a function mean in-cloud vertical velocity for each column from a single time step at 8 h. Marker colour is based on vertically integrated INP mass mixing ratio (the total INP mass above $1\,\mathrm{m}^2$ of sea surface area). Simulations are with (left) and without (right) marine INPs and high background INP concentration.

Higher cloud droplet freezing rates are related to updraughts (positive vertical velocity) and marker colour shows that the updraughts have higher INP mass concentrations. INP concentrations range from less than $5\,\mathrm{\mu g\,m^{-2}}$ to above $14\,\mathrm{\mu g\,m^{-2}}$, so the variability is as high as $\pm 50\,\%$ compared to the $9.6\,\mathrm{\mu g\,m^{-2}}$ background INP concentration. Nucleation rate and INP mass are also linked so that higher INP mass (more larger particles that can actually freeze) leads to higher nucleation rate at a constant
vertical velocity. The brief explanation for these findings is related to the vertical transport of INPs. Unfrozen aerosol-phase INPs have high concentration near surface (see Fig. 5) mainly due to ice crystal sublimation. Updraughts bring these INPs to cloud regions where they become cloud droplets which continue rising until they reach temperatures low enough for freezing. Stronger updraughts can reach lower temperatures just below inversion layer, which means higher freezing rates. Freezing rates decrease when the most effective INPs have been frozen. Ice crystal growth is not dependent on vertical velocity, but
downdraughts increase sedimentation rates, which reduce INP concentrations. The strongest downdraughts also originate from the cloud top region which is depleted from INPs (Fig. 5) due to ice crystal sedimentation.

### 3.5 Sensitivity tests

Here we examine the effects of microphysical (aerosol and cloud droplet sedimentation, and immersion freezing of unactivated aerosol) and meteorological (de-coupled marine boundary layer) model considerations mentioned above. Figure 9 shows the
test simulations divided into two groups according to the reference case. The reference case for the microphysical tests (enabled aerosol and cloud droplet sedimentation or aerosol freezing) is the simulation with high background INP concentration and marine INP emissions on. The effect of de-coupled marine boundary layer (MBL) is tested by running simulations based on

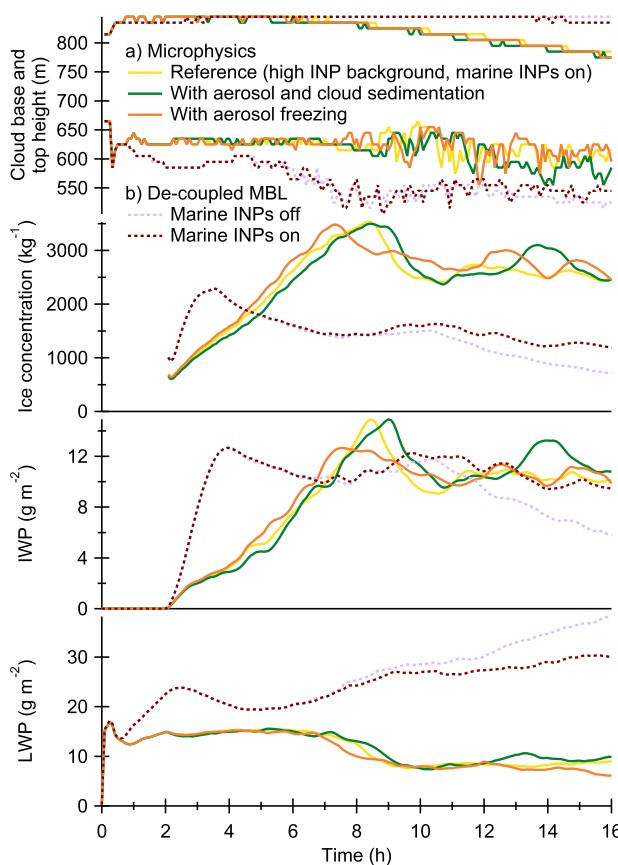

**Figure 9.** Sensitivity tests related to a) microphysics (solid lines) and b) de-coupled marine boundary layer (dashed lines). The microphysics tests include aerosol and cloud droplet sedimentation or aerosol freezing switched on when compared with the reference case where they are disabled. The de-coupled marine boundary layer (MBL) tests incudes simulations with modified initial temperature and humidity profiles when marine INP emissions are off or on.

the modified initial temperature and humidity profiles, which are explained below, when marine INP emissions are off or on (high background INP concentration for both simulations).

Allowing aerosol and cloud droplet sedimentation (disabled in the default simulations) has two potential effects on INPs. First, aerosol sedimentation could bring INPs from the free troposphere or remove those from the near surface layer by dry deposition. Second, sedimentation could have an impact on vertical distributions. Cloud droplet sedimentation redistributes cloud water, which influences clouds as explained in Ovchinnikov et al. (2014). Due to this side effect on clouds, it is not possible to fully isolate the effect of sedimentation on INPs. However, simulations made with and without aerosol and cloud

droplet sedimentation show negligible differences. Advection and ice crystal sedimentation dominate vertical mixing, and the slow removal of INPs by dry deposition is almost fully compensated by a flux of INPs coming from the free troposphere.

Figure 9 shows that allowing aerosol immersion freezing (freezing in the default simulations is limited to cloud droplets) has a small impact on ice crystal number concentration. The main reason for this is the fact that freezing is practically limited to the cloudy regions (sub-saturated regions are too warm) where the aerosol can freeze before or after cloud activation. Aerosol freezing rate is about 10 % of the total freezing rate, so it is not insignificant. However, due to the above-mentioned reason, cloud droplet freezing rate is reduced by the same amount so that the total freezing rate is about the same as that in the simulation without aerosol freezing (Fig. 6). Most of the aerosol freezing takes place at the top of cloud. This is not related to downdraughts (or subsidence) bringing new INPs from the free troposphere. Instead, spatial correlation between cloud activation/de-activation and aerosol freezing rates at the cloud top indicates that temperature fluctuations, for example due to radiative cooling or mixing, first cause the release of INPs from evaporating cloud droplets and later initiate cloud activation and aerosol freezing. In the latter case, the instantaneous aerosol freezing can take place before the diffusion limited droplet growth leads to cloud activation.

Meteorological conditions are crucially important for clouds, but here we focus on the one that has direct relevance for the vertical transport of marine INP emissions, namely de-coupled marine boundary layer. The original ISDAC simulations were initialized with a de-coupled marine boundary layer, which reduces vertical mixing and partially isolates the near surface layer from the rest of the boundary layer (Wood, 2012). Figure 9 shows simulations with and without marine INP emissions (both with high concentrations of background INPs) when the boundary layer is de-coupled and humid as in the original ISDAC case study (Ovchinnikov et al., 2014). Temperature is increasing $0.004\,\mathrm{K\,m^{-1}}$ and total water mixing ratio is decreasing $0.00075\,\mathrm{g\,kg^{-1}\,m^{-1}}$ within the lowest $400\,\mathrm{m}$. Due to the different heat and humidity contents, cloud states are different for coupled and de-coupled boundary layers. Comparing simulations made with and without marine INP emissions when boundary layer is de-coupled shows that marine INP emissions have negligible effect at least during the first 10 hours. This shows that de-coupling can indeed prevent marine INPs reaching clouds. After the first 10 hours, boundary layer becomes more coupled, so marine INP emissions start to influence ice crystal number concentrations.

## 4   Conclusions

In this study we examined the potential effects of marine Ice-Nucleating Particles (INPs) on shallow mixed-phase clouds by using a large eddy simulator UCLALES-SALSA (Tonttila et al., 2017; Ahola et al., 2020), which has prognostic aerosol, cloud and ice phase INP size distributions. Simulations were made by adjusting initial background INP concentrations and INP emissions with sea spray so that reasonable cloud ice crystal number concentrations were seen for a wide range of source strengths. Our simulations show that in the case of well mixed (coupled) boundary layer, updraughts are efficient in transporting marine INPs up to the clouds where droplet freezing can take place. When the background INP concentration is low, which means that free troposphere is not a significant INP source, relatively low marine INP emissions can maintain the simulated mixed-phase clouds. While the free troposphere is separated from the clouds by an inversion layer, which reduces vertical mixing, marine INPs are emitted directly to the boundary layer.

Our simulations with UCLALES-SALSA support the previous findings about the importance of INP recycling (Fan et al., 2009; Solomon et al., 2015). This means that the smallest precipitating ice crystals lose all ice in the near surface sublimation layer so that the original INPs are released. Updraughts transport these INPs, as well as those from the sea spray aerosol, back to the clouds where they may again initiate droplet freezing. Detailed examination of the 3D model outputs show that ice nucleation is focused on the updraught regions and these regions may have up to 50 % more INP mass compared with the background INP mass. On the other hand, downdraughts are depleted by up to 50 %, because they originate from the cloud top where ice crystal sedimentation reduces INP concentrations.

Prognostic ice microphysics including explicitly modelled ice nucleation is important for the simulations as this allows feedbacks between INPs and clouds (Paukert and Hoose, 2014; Savre and Ekman, 2015; Ahola et al., 2020). Precipitation removal is the most important feedback in our simulations. Increasing INP concentration increases precipitation removal rates, so most of our simulations ended up having similar cloud ice contents. Precipitation feedback also prevents complete glaciation, which happens in the case of fixed ice crystal number concentration (Ahola et al., 2020).

Efficient INP recycling, feedbacks between INP emissions and precipitation removal, and the fact that marine INPs are emitted directly to the mixed layer mean that modest marine INP emissions can maintain mixed-phase clouds at least in the conditions used in our simulations. Although significant uncertainties are still related to ambient INP emissions, our simulations support the current view (Vergara-Temprado et al., 2017; Huang et al., 2018; McCluskey et al., 2019; Zhao et al., 2021) that marine INPs can have a dominant role in remote regions far from continental dust sources.

*Code and data availability.* The source code of UCLALES-SALSA version used in this work is available from https://github.com/UCLALES-SALSA/UCLALES-SALSA/tree/isdac_poa (last access 3 March 2021). Brief description of the simulations and the data used in this publication are available at https://doi.org/10.23728/FMI-B2SHARE.752699D19F34489BBDAEAD7E7C591E27 (Raatikainen, 2021, Last access 11.11.2021).

*Author contributions.* TR designed and performed the model simulations. All authors have contributed to developing the UCLALES-SALSA model. TR prepared the manuscript with contributions from all co-authors.

*Competing interests.* The authors declare that they have no conflict of interest.

*Acknowledgements.* This research has been supported by the Academy of Finland (grant nos. 322532 and 309127) and Horizon 2020 Research and Innovation Programme (grant no. 821205). The authors wish to acknowledge CSC – IT Center for Science, Finland, for computational resources.

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
