# Peer review of "The effect of marine ice-nucleating particles on mixed-phase clouds"

_Atmospheric Chemistry and Physics, 2021_

## Author Comment (AC1)

We would like to thank both referees for their useful comments, which have helped us to improve the manuscript. Below are the original referee comments (*shown with italicized font*) and our replies are below each comment (shown with blue font). Some specific updates to the manuscript can be found after our replies (deleted, added and unchanged text). In addition to the updates suggested by the referees, we fixed a few typos and Fig. 7 where one dark blue line should have been black, and we updated the reference to our simulation data which was moved to a permanent research data repository.

**Response to Referee 1**

*This manuscript describes a very interesting experiment carried out using high-resolution large eddy simulations to study the effect of INP concentrations in mixed-phase clouds and particularly the role of marine INPs. The study focuses around different cases with different background INP concentrations, switching on and off the marine INP emissions. The results are relevant and interesting, highlighting the effect of marine INPs under certain conditions (related to the background INP and the meteorological conditions). The study also highlights the importance of INP recycling. However, some important issues need to be addressed. I would recommend the manuscript for publication after these issues have been resolved.*

*One of the major points to address is the fact that the methods section is difficult to follow. This is because it is a bit unorganised (e.g., the INP description is scattered through the text). Additionally, there are many things that need to be described much more precisely, since they are relevant for the results (particularly the ice-nucleation scheme, the calculations of INPs from aerosol particles and the studied area). The fact that this section is complicated and a lot information is missing might have affected my understanding of other parts of the manuscript. I suggest the authors spend some time improving this section substantially (see specific comments below).*

Method section is now reorganized and more detailed (details below). Please see the revised manuscript for the updates.

*Another important point I see in this manuscript is the fact that some processes appear to be disconnected from existing measurements. This is particularly the case of INPs concentrations. I understand it is difficult to do so, given the fact that model assumes a temperature independent INP concentration, which then might or might not act as an INP depending on its size and temperature. Most measurements of INP use a singular description of the process and therefore produce INP vs temperature spectrums, which might be difficult to compare to the concentrations reported here. However, I still think substantial effort needs to be done in order to address this. Are the simulated background and marine INP concentrations realistic? Without these comparisons, one could argue that maybe the modelled marine INP component is too high compared with the background (or the opposite). Realistic INP concentrations are necessary to support one of the main conclusions of the study (marine INPs have an influence on mixed-phase clouds).*

As the referee points out, direct comparison between modelled and observed INPs is difficult. However, we have added a description of typically observed INP concentrations (at 268 K) to the updated Introduction and conduct a comparison to ISDAC observations in the updated Methods section. The comparison shows that the highest INP concentration in our model (RH=100%, T=268 K, and a residence time of 10 s) is consistent with the observed INP concentrations. In the actual

simulations, we use a range of INP concentrations lower than this, which should cover the observed concentration range.

We cannot compare our marine INP emissions to observations or other parameterizations representing continuous emissions because emissions used here are to some degree higher to have impact during the typical length of a LES simulation, which is shorter than could be the case for real clouds that might develop over several days. In the end, our marine emissions seem to have a reasonable impact on clouds (small when background INPs dominate, and large enough in the absence of background INPs), which is what we aimed for. For this purpose, we added a short review of recent modelling studies about the marine INP sources to the Introduction. Please see the revised manuscript for the updates.

**Specific comments**

**Introduction**

*General: As a suggestion, I do not think it is necessary to add e.g. before the references.*

We have removed most of those here and elsewhere.

*Lines 21-28: The first sentence refers to immersion mode, while the rest of the paragraph defines the ice-nucleation modes, finishing with the immersion one. I suggest altering the order of the paragraph: starting with a description of the modes, which finishes with the immersion mode. Then add the statements in how immersion mode affects shallow mixed-phase clouds.*

The order changed as suggested.

*Line 33: The mentioned reference only provides INP vs temperature data. I would add other examples of fieldwork where INP vs temperature and INP vs relative humidity are shown.*

We removed reference to humidity, because our focus is on mixed-phase clouds and immersion freezing where relative humidity is about 100 %.

*Line 35: I suggest not including soot as one of the most important types of INPs. Although this might be the case for deposition mode, there is not much evidence that it is for the immersion mode and these part of the introduction (and the manuscript in general) is referring to the immersion mode.*

This is true, so we removed soot.

*Line 38 "In the absence of dust": I suggest mentioning that there some dust sources in cold high-latitude environments (Bullard et al., 2016) and they could be important local INP sources in the absence of dust from deserts too (Tobo et al., 2019, Sanchez-Marroquin et al., 2020).*

This sentence was referring to the remote marine regions far from the continental dust sources, which would otherwise dominate. We have clarified this paragraph and also mention the high-latitude dust/INP sources based on the three references (Bullard et al., 2016; Tobo et al., 2019; Sanchez-Marroquin et al., 2020). Please see the revised manuscript for updates.

**Methods**

*Sect. 2.2: Have Secondary Ice Processes been implemented?*

Secondary ice processes were left out of the scope of the current study, as the cloud conditions during ISDAC do not favor secondary ice processes. For example, ice crystals are pristine dendrites

and cloud temperature is less than 263 K. This is now mentioned in the manuscript (Sect. 2.1 in the revised manuscript).

*Line 65: A bit more of information on the ISDAC should be added; where and when it happened, the type of measurements that were conducted that are used in this work.*

Information about ISDAC has been added to the upgraded Methods section (Sect. 2.1). We also clarify that we are not using ISDAC measurements directly but some of our model inputs are from a LES study based on ISDAC.

*Line 70: Is the 4 L$^{-1}$ concentration based in something? If this comes from a measurement of the ISDAC campaign, it should be stated.*

It is an observation-based upper concentration limit used in the previous LES study. The description of ISDAC observations and how they relate to ours and the previous LES simulations is now revised.

*Line 77: I suggest adding a full description of the species used for this work. I would also add a description of the bin range.*

This description is revised so that only the species used in this work are mentioned (Sect. 2.1). Water is the substance that partitions between vapor and condensed phases, dust is the insoluble ice-nucleating material, and sulphate is a soluble substance. The species have essentially predefined physical properties, so we clarified the changes from the default values (sulphate properties changed from sulfuric acid to ammonium bisulfate). The bin range is also described (the beginning of Sect. 2.2).

*Line 83: It is necessary to add a description of the ice-nucleation scheme and how it relates to dust and sea spray. Is the scheme (and its output) consistent with recent experimental INP measurements?*

This part of the text is updated so that the key parts of the ice-nucleation scheme including the details about background and marine INPs are now described in the updated Methods section (Sect. 2.2). The equations can be found from our previous publications (Ahola et al., 2020). We are comparing our INP concentrations to the ISDAC observations and they seem to be consistent. We are also comparing our INP concentrations to those from different measurements and model simulations (overviews given in the updated Introduction). More detailed comparisons are not useful, because our focus is on cloud state rather than ice nucleation parameterizations. In the revised manuscript we clarify that our approach in this study is to vary the INP concentrations in the range that produces reasonable mixed-phase cloud while keeping the ice nucleation parameters constant. This approach was selected because we cannot adjust marine and background aerosol ice nucleation parameters independently. Also, this approach allows us to simulate different INP concentrations covering the range of typical observations.

We will also clarify our terminology regarding INPs, which seem to have caused confusion (Introduction and Sect 2.2). Experimental studies relate INP concentration to those particles that freeze at given instrument conditions (mainly temperature, RH, residence time and detection limits). This is not practical for modelling studies where droplet freezing depends on simulated conditions and particle properties. Also, any dust-containing particle (marine or background) in our stochastic ice nucleation scheme has a non-zero freezing probability although this can be extremely low or high. For this reason, INP in our simulations is a synonym for a dust-containing particle that can freeze via the immersion freezing mechanism.

*Line 91 "adjusting the initial concentrations of INPs": How was this done? Is this validated based on any experimental study?*

Concentration was adjusted by changing the fraction of externally mixed INPs in the initial aerosol size distribution. This is now clarified in the revised manuscript. A brief comparison to observations has been added (near the end of Sect. 2.2).

*Line 92 "ammonium bisulphate" (and in general through the whole manuscript): Is the ammonium bisulphate an immersion mode INP or is it just there to activate the dust as a CCN prior to ice-nucleation? The later does not seem the case, since later the authors state that the dust and ammonium bisulphate are externally mixed. I do not think ammonium bisulphate is a relevant type of INP in the immersion mode, according to the literature (it could be for cirrus clouds but the focus here is mixed-phase clouds). The inclusion of this species to the INP population needs to be much better explained and justified (there are not even mentions to it in the introduction but then it seems to be a major thing for this work) or removed.*

Ammonium bisulphate is there just to activate the dust. Insoluble dust particles would not produce liquid droplets or activate without a soluble coating as we are not accounting for water adsorption on solid surfaces, which could lead to activation of largest dust particles. We clarify that INPs (=dust + ammonium bisulphate) and the remaining aerosol (=ammonium bisulphate) are externally mixed. We have also clarified our terminology and this section in general.

*Figure 1. As a suggestion, I would use μm instead of m for consistency with most of the aerosol studies.*

Units fixed.

*Line 95 "(set to 0.00015)": Does this mean that the INP concentration is 0.00015 times the dust concentration, regardless of the temperature? In general, this point would be addressed by including a much more detailed description of the ice-nucleation scheme, and if it is linked or justified based on any experimental study.*

The fraction of dust-containing particles (=INPs) from total aerosol is 0.00015, and this is independent of temperature. We have clarified the description of the ice-nucleation scheme and our terminology. A brief comparison with observations is also given.

*Line 108 "Ice nucleation follows the approach used in Ahola et al. (2020)". The description of the ice-nucleation scheme seems to be split between the lines 85 to 97 and 108 to 120. I suggest merging this. Additionally, as previously explained, the description needs to be much more extensive, clear and organised.*

The description of the ice nucleation scheme is now revised (see Sect. 2.2 in the updated manuscript).

*Line 115 "but marine and background INPs become internally mixed": I do not really understand this part, please explain it better. If ammonium bisulphate can be excluded as an immersion mode INP, maybe you can have the background and marine INPs externally mixed?*

SALSA can have two externally mixed particle modes and one of them is reserved for the non-INP aerosol (ammonium bisulphate from background and sea spray), so there cannot be separate modes for the marine INPs and background INPs. We have clarified our terminology and this section especially.

*Line 117 "internally mixed dust and ammonium": Isn't this externally mixed as stated in line 114?*

There is just one particle mode for the two INP types (see the reply above), so these particles become (internally) mixed. We have clarified this in the updated manuscript.

*Line 121-128: How is the SSA related to INP? Is it based in any experimental parameterization? Is there any temperature dependence? How do the modelled INP concentrations compare with available measurements of INP or concentrations from other models?*

The SSA emission parametrization is based on observations, but our marine INP emissions are based more on simulations as explained above and in the updated Methods-section. We cannot use existing marine INP parameterization aimed for global simulations, because such parameterizations would need much more time to produce non-negligible INP concentration than is possible with the short LES simulations. These parametrizations also require information about the marine biological activity, which is not included in the LES. For this reason, we adjusted marine emissions so that we see a small effect on clouds with the highest background concentration (low marine contribution in the polluted case) and the marine emissions are able to maintain mixed-phase cloud in the absence of background INPs (marine-dominated in the clean case). The updated Methods-section contains a comparison with observations as explained above.

*Line 139 "adjusting the fractions of INPs". Explain this and if the adjusted INP concentrations are comparable to measurements.*

We have clarified our methods. Observations are also briefly compared.

**Results and discussion**

*Figure 2 (legend): I suggest using only 5 colours for this graph, each of them corresponding to each INP fraction, and then use the "dashed" "not dashed" to refer to Marine emissions on and off.*

This really improved the figure, so we did this modification here and elsewhere in the manuscript.

*Line 169 "Simulations where marine INP emissions are switched off are excluded for clarity": Why, isn't this comparison interesting too?*

There isn't much difference between simulations where marine emissions are on or off, but certainly we can show both cases (also in Figs 4 and 6).

*Figures 3 and 6 would benefit from having a title specifying that each column refer to mass and number respectively.*

Titles added.

*Lines 182 to 186: From what I understand, this analysis is done on what the authors call "INP" which is not the particles that nucleate ice but a temperature independent fraction of the total aerosol (background and marine). Then, some of these "INPs" will act as an INP based on their size and temperature. Hence, this analysis of INP budget, seems more like and aerosol budget analysis. I think the authors should justify why this analysis is relevant to the particles that will nucleate ice in the model, which are likely the ones that are numerous and carry enough surface area (see next comment). As previously mentioned, a more concise, systematic and expanded description of the INP scheme would help understanding this too.*

As explained above, INP in our simulation is a particle that can nucleate ice, because we cannot identify those particles that will eventually freeze as it depends on time exposed to the modelled

temperature and humidity. We have clarified this and expanded the description of the ice nucleation scheme.

The INP budget analysis is not an aerosol budget analysis, because only a small fraction of the aerosol are INPs (i.e., contain dust). By accounting for all INPs (dust-containing particles), we can have a closure. On the other hand, if only those particles that freeze at certain temperature and humidity over a fixed residence time would be counted as INPs, the INP budget analysis would fail, because the model-predicted freezing would not match with this definition of INP.

We prefer to use dust mass in the analysis as it is related to the largest particles, which are more likely to nucleate ice. The number analysis focuses on particles larger than 159 nm, which is the approximate limit based on simulations. Reply to the question about surface area is given below.

*Line 186 "we will focus more on the INP mass": What about INP surface area? Your mass INP will be dominated by the upper end of the aerosol range, however, aerosol particles of medium sizes would carry a substantial amount of surface area (if not the majority) and they might not contribute much to the mass or number, biasing the results. This seems important since I guess the stochastic ice-nucleation scheme will be dependent on the surface area of the aerosol particles? I suggest addressing this too, or justifying why surface area has been excluded of the analysis.*

Surface area has been excluded from the analysis, because it is not included in the model outputs, which describe number and mass statistics. These are the model outputs and shown in the manuscript, because number has its well-known common usages and mass has special importance for budgets as it is always conserved. It is true that surface area influences ice nucleation, but not as directly as one could think. Specifically, the area in our stochastic ice-nucleation scheme is not related to ice-active surface site densities reported in several experimental studies. Since the number and mass are clearly needed, we will keep using them.

*Section 3.2 (general): This comment is related to previous comments about comparison with observations. I think more effort should be done in order to show that the modelled INP budgets are realistic, showing some consistency with measurements (I am aware there might not be measurements at that specific time and location, but the concentrations could be compared to the closest available observations). I appreciate it is difficult to perform this comparison, since most measurements are carried out using a singular description of ice-nucleation while this work uses a stochastic one. However, I still think some effort should be put into this.*

We added a brief comparison to the Methods-section. There we simulate the number of ice crystals formed from our initial aerosol size distribution based on our cloud top temperature, RH and a typical instrument residence time (10 s). This INP concentration seem to be consistent with ISDAC observations and typical observations of continentally influenced air masses elsewhere. As explained above, there are no suitable observations for marine INP emissions, so their emissions are adjusted based on their relative effects on clouds.

*Line 203 "over simulation time 7-8h": Make clear than you are doing a one hour average starting at the seventh hour of the simulation. It took me a while to realise what the authors mean.*

We have clarified this here (see below) and elsewhere.

"The profiles are  one-hour averages over the eighth simulation hour of the instantaneous model outputs…"

*Line 205 "The other simulations show": Does this mean the same analysis applied in other 1 hour time intervals? Are all the conclusions of this analysis valid for the other time intervals? This should be explained more concisely.*

First, we clarify that the 1-hour averaging is just to reduce noise and fluctuations seen in instantaneous profiles (see below). With the other simulations we mean those with different background concentrations and with and without marine emissions. Different time periods are also included. The main driver of the processes shown in Fig. 5 is precipitation, which influences vertical INP distributions, and precipitation can be related to ice crystal number concentration. This brief explanation should become clear after reading the following paragraphs where we explain Fig. 5.

"The purpose of the averaging is to reduce noise and fluctuations related to the instantaneous model outputs. The time interval was selected as an example, because the highest ice number concentrations are seen at that time just before precipitation rates increase significantly. The other  time periods and simulations (different background INP concentrations and with and without marine INP emissions) show similar behaviour, but magnitudes of these processes depend mostly on ice crystal number concentration, which influences precipitation and eventually vertical INP distributions (see the description below)"

*Line 237 " Simulations where marine INP emissions are switched off are excluded for clarity": Why? Isn't it making any difference?*

Clarity was the original reason, but we have now added the simulations without marine emissions.

*Line 265 "ice nucleation rates": Could you describe what this magnitude means in the model? Is it the number of primary ice freezing events per second and per surface area of INPs?*

It is the number of primary freezing events per second in a column of air above 1 m² of sea surface area. This is now clarified in the manuscript:

"Figure 8 shows vertically integrated ice nucleation rates (the number of primary freezing events per second in a column of air above 1 m² of sea surface area) as a function of mean in-cloud vertical velocity for each column of the domain from a single time step at 8 h (4096 columns in total). Marker colour is based on the corresponding vertically integrated column INP mass mixing ratio (the total INP mass in aerosol, cloud droplets and ice crystals above 1 m² of sea surface area)."

*Figure 8 would benefit from having a title indicating which panel refers to marine missions on an off.*

Titles added.

*Line 271 "updraughts have higher INP mass concentrations": This section is interesting. However, I find the magnitude INP mass in a column a bit disconnected. Wouldn't it be better to give this in INP number?*

As explained in the previous replies, INP number in our terminology includes all particles that can freeze, but only a fraction of those will be frozen in the simulated conditions. The number is dominated by small INPs (especially marine emissions), which won't freeze in the current conditions. That is why we prefer to use mass as it is representative of the largest particles and therefore has similar magnitudes for the simulations with and without marine emissions.

*Line 282: "aerosol freezing": Is this what other studies refer to as deposition (or pore condensation freezing) ice-nucleation? It is not clear. If so, please indicate in the method section how this process is parameterized. If my assumption is right, link it with the existing literature, which suggests the same (this process is not very important for mixed-phase clouds, when comparing with immersion freezing).*

Here (and elsewhere in the manuscript) aerosol freezing means immersion freezing of aqueous INP aerosol. The mechanism is exactly the same as that for cloud droplets, but is related to interstitial aerosol and aerosol outside the cloud. This is clarified in the updated manuscript:

"Here we examine the effects of microphysical (aerosol and cloud droplet sedimentation, and aerosol freezing immersion freezing of unactivated aerosol) and meteorological (de-coupled marine boundary layer) model considerations mentioned above."

***Conclusions:***

*Line 323 "which has prognostic aerosol, cloud and ice phase INP size distributions". This sentence is describing the methods; it would go better in the first paragraph of the conclusions.*

We moved this to the first paragraph.

***Other***

*Ice nucleation: Ice-nucleation*

Typically ice nucleation alone is written without hyphen, so we follow this convention.

*Line 3 "ice nucleating particles (INPs)": Ice-Nucleating Particles (INPs)*

Done here and elsewhere. The hyphen can be used in this context.

*Line 88 "258 K or -15 °C": I suggest sticking to one temperature unit through the text.*

We removed temperatures given in degrees Celsius.

**Response to Referee 2**

*In the paper "The effect of marine ice nucleating particles on mixed-phase clouds" by Raatikainen et al. 2021 the authors present a case study with the large eddy simulation model UCLALES-SALSA on the importance of local marine INP emissions on shallow marine mixed-phase clouds. They show that marine aerosol emissions can contribute to the boundary layer INP budget and influence the cloud (differently strong depending on background conditions). The study is very interesting and the analysis done very carefully. It fits well to ACP. Some of the background, basic assumptions and limitations of the study are unclear and have to be phrased out, therefore I suggest to accept the paper after major revisions.*

***General comments:***

*- The (regional) focus of the paper is unclear. In the introduction a lot of the literature etc. is focused on the Southern ocean, but the case study presented in an Arctic case study. Both regions are regions where marine aerosols as INP could play a role, but the introduction could be clearer and/or less diverse and more focused. The choice of ISDAC as a case study could be set in a better context.*

We don't have specific regional focus. However, ISDAC setup was selected as it is used in different model sensitivity studies and provides a well characterized setup. Basically, our simulations could represent mixed-phase clouds anywhere in the high latitudes. Although the ISDAC aerosol concentrations are representative of continentally influence air masses, we adjust INP concentrations so that they represent cleaner regions. The updated Introductions gives an overview of the marine mixed-phase clouds while the updated Methods-section focuses more on the ISDAC case study and our model inputs.

*- Another aspect related to the choice of ISDAC and the setup of the study is the question how valid it is to assume that sea spray emission in the ISDAC region acts like on "open ocean". The sea surface of ISDAC was not covered by an area-wide ice cover, but sea ice was present which certainly has an influence on sea spray emission. This should be discussed more in the manuscript. It should be stated how large the domain was (maybe with some map as well), how/where it was ice covered and if that was taken into account when calculating the sea spray flux. For other Arctic case studies there might not be a marine source of aerosols from the surface just because of the presence of an ice cover.*

As mentioned above, we are not focusing on replicating the ISDAC observations but take some of the LES inputs from the previous case studies. Because we are not focusing on the effect of ice cover on marine emissions, which could be a good topic for another study, we ignored the effect sea ice. This has no impact on our simulations as sea-surface temperature at the freezing point and negligible surface fluxes are valid assumptions with or without sea-ice. We also assume uniform ocean surface. Domain size is now specified in the manuscript, but without a map, because our LES simulations do not depend on the specific location.

*- The description of the ice nucleation scheme used is very scattered throughout the text and difficult to follow. It seems that a CNT approach is used and in the text it is mentioned that the contact angle (for which species?) was increased because of the warm temperatures of the case study. It is unclear if that only yields for the dust aerosols and what the change of contact angle does represent (since that necessarily reflect on the composition/population of aerosols). It is not explained if the same parametrisation is used for the sea spray aerosol or not. If in the presented model setup sea spray freezes with the same contact angle as the dust particles (or even the adjusted contact angle) this would be quite a limitation on the study when it comes to the interpretation of the results. Sea spray does not trigger freezing as effectively as dust and the whole conclusion (that sea spray emission can*

*have a significant influence on clouds) depends quite a lot on the assumptions made for the freezing of both aerosol types.*

We have clarified the Methods section in the upgraded manuscript so that it contains more detailed description of the ice nucleation scheme. We also clarify that we adjust background and marine INP concentrations independently while assuming the same constant ice nucleation parameters and fixed INPs size distributions so that reasonable mixed-phase cloud is simulated. This is partly due to a model limitation, which forces us to use the same the ice-nucleation active species (dust) for both background and marine INPs. This means that the contact angle is the same for both INP types (or the dust that they contain), so we cannot adjust contact angles independently, but we can adjust marine and background INP concentrations. Sea-spray (or the background) aerosol without the dust is not freezing in our simulations. We have clarified our manuscript so that marine emissions are changed to marine **INP** emissions (also in the case of background **INPs**).

As explained in the updated manuscript, current contact angle represents dust or any other equally active INP. Marine INPs could be more effective, but the absolute values are unknown. The current contact angle was selected so that about half of the INPs (dust-containing particles) freeze in our simulations. Using a different contact angle in our simulations would be possible, but it wouldn't influence our conclusions, because we would have just adjusted the INP concentration so that about the same number of particles would freeze in our simulations.

***Specific comments:***

*- There is at least two studies missing when it comes to the simulation of marine aerosols as INP: Huang et al., Global relevance of marine organic aerosol as ice nucleating particles, ACP 2018 (https://doi.org/10.5194/acp-18-11423-2018) and McCluskey et al., Numerical Representations of Marine Ice-Nucleating Particles in Remote Marine Environments Evaluated Against Observations, GRL 2019 (https://doi.org/10.1029/2018GL081861). Check again your background literature.*

Both studies were mentioned in the manuscript, but McCluskey et al. (2019) was not mentioned in the introduction. The reference is now added among two new ones (Wex et al., 2019, and Hartmann et al., 2021). For updates, please see the updated manuscript.

*- p. 2, l. 69-71: I would suggest to remove the last two sentences on the case study here. It is not the approach used in the paper and thus a bit confusing.*

This paragraph is revised (please see the updated manuscript).

*- p. 6, l. 155-156: I don't understand what is meant by "cloud base and top heights represent the domain minimum/maximum value". Are the domain minimum/maximum values used for the cloud base and clou top heights here? What was the range of these values throughout the domain?*

Cloud base and top height are the domain minimum base and maximum top heights, respectively, so they represent the full extent of the cloud deck. This is now clarified in the manuscript (see below). This seems to be the most common definition at least in the LES modelling community.

"Results from our simulations are shown in Fig. 2. Cloud base and top heights  are the domain minimum base and maximum top heights, respectively, so they represent the full extent of the cloud deck."

The range of cloud base and top heights depends on time and simulation. For the simulation with high background and marine emissions on, the base heights range from 615 m to 715 m and the top

heights range from 805 m to 835 m (simulation time 8 h). This represents a high variability based on liquid water path standard deviation (diagnostic model output, which is not shown).

*- p. 6, l. 164: That sounds misleading, the simulation INP on with lower BKGD is similar to BKGD medium without marine sources, but not the simulation with zero BKGD? At least not until 14 h (and there most of the simulations look the same). So it does not seam correct to say it is the marine emissions alone.*

We will clarify that the largest differences between simulations with and without marine emissions are seen with the lowest background concentrations (see below). This means that the background aerosol has a buffering effect. We will clarify that marine INP missions alone will lead to the same final state as the medium (or higher) background without marine INPs.

"Marine INP emissions The largest differences between simulations with marine INP emissions on or off are seen with the lowest background INP concentrations. This means that marine INP emissions become more important with decreasing background INP concentration. In factaddition, marine INP emissions alone seem to have the same effect produce the same final mixed-phase cloud state as having the medium or high INP background without marine INP emissions."

*- p. 7, l. 180: "The INP mass includes the total dust mass in aerosol, cloud droplets and ice crystals." How exactly is that done/meant? Is it accounted for which particles were activated/froze and how is that done for the different categories (when looking at the total dust mass it is not known how the ratio is of particles that would lead to freezing; when looking at the dust aerosols within ice crystals, they did obviously nucleate ice)?*

This is the sum of dust (the modelled INP in the immersion freezing parameterization) mass in aerosol, cloud droplets and ice crystals (the update shown below). In that way the sum of the different terms determines whether the total INP mass concentration decreases or increases. While precipitation and surface emissions are related to ice and aerosol, respectively, subsidence will influence all species. Because INPs will encounter different conditions during their life cycle, we cannot know if they will be frozen at some point (only the dust in ice crystals is known). That's why we prefer using mass as it is related to the largest and most efficient INPs. We have clarified our terminology regarding INP (all particles that can initiate freezing, i.e., dust-containing particles), which differs from that used by experimentalists (particles that initiate freezing at given instrument conditions).

"The INP mass includes the total dust mass in mass of the insoluble ice-nucleating material (represented by dust in the model simulations) in aerosol, cloud droplets and ice crystals."

*- p. 8, l. 185: What is meant by "significant fraction of ice in all our simulations"?*

This number is based on the average 10$^{th}$ percentile of the ice crystal number size distribution (see the figure below). In practice, this is the INP dry size so that 90 % of the ice crystals originate from particles larger than this. The closest bin limits near this range are 95 nm, 159 nm and 266 nm, so we decided to take the 159 nm as our definition. Because this limit is time and simulation dependent, we will focus more on mass and keep this discussion brief. We mention the 90 % limit in the updated manuscript:

"This is the lower limit of the first size bin (Fig. 1) that has a significant fraction of ice in all our simulations. Specifically, at least 90% of the ice crystals originate from INPs larger than 159 nm during the first eight simulation hours."

[Figure]

*- p. 8, l. 188: Why is the surface INP a 0.005 fraction of total SSA flux which is a lot higher than fraction of initial aerosol concentration?*

The fractions are not comparable, because the surface emission fraction is related to the flux of particles starting after the one-hour spin-up while the background fraction determines the initial INP population for the whole domain as a fraction of the initial aerosol. Initial aerosol concentration was relatively high, so a lower fraction is needed. On the other hand, the SSA particles are relatively small (see Fig. 1 in the manuscript), so they are less effective INPs.

*- p. 9, l. 205: Specify: the other simulations (context unclear?).*

This refers to the other simulations with different background INP concentrations and marine emissions on or off. We have clarified this in the revised manuscript:

"The other  time periods and simulations (different background INP concentrations and with and without marine INP emissions) show similar behaviour, but magnitudes of these processes depend mostly on ice crystal number concentration, which influences precipitation and eventually vertical INP distributions (see the description below)"

*- Fig. 5: Why is the simulation set with high background concentration chosen for this figure? Would it not make more sense to choose a lower background concentration simulation set?*

Different simulations (and time periods) were considered, but we selected this case where the background and marine simulations have similar ice crystal number concentrations and the INP concentrations are the largest. The high concentration is generally interesting because it is close to the limit where the cloud glaciates. In addition, the high concentration means reduced noise and fluctuations so that the profiles look smooth. When the background and marine simulations produce similar ice crystal number concentrations, also the profiles have similar magnitudes. Therefore, we can show the different profiles from the two simulations in single figure, which clearly visualizes the differences and similarities between the simulations. For consistency, we show this same case in Figs 7, 8 and 9.

*- Fig. 5 caption: Rephrase: "diffusion, subsidence and surface flux contribution" instead of "diffusion, subsidence and surface" (?).*

Done.

*- INP and ice budget: Here it could be beneficial to have the plots from both analysis next to each other and maybe combine analysis in one text section (only a suggestion, also fine to leave it as is).*

This could be possible, but it would lead to a long section. Also, parameters show in Figs 3 and 6, as an example, are mostly different so showing them next to each other could cause confusion. So, we prefer to keep the separate sections.

*- Fig. 8: It would be interesting to plot an equivalent figure as a function of temperature.*

The figure below shows the ice nucleation rate as a function of mean in-cloud temperature (marine emissions on and high background INP concentration). There isn't that much variability between column mean temperatures (or in-cloud grid cell temperatures), so INP concentrations and vertical fluxes are more important for the nucleation rate. That is why these are shown in the manuscript instead of the temperature dependence. In general, our ice nucleation scheme was adjusted to produce reasonable ice crystal number concentrations for the current cloud case, so it cannot be used to predict INP concentration for other temperatures.

[Figure]

*- p. 13, l. 273: "higher INP mass leads to higher nucleation rate at a constant rate of vertical velocity"*
*- is it not the other way around (higher nucleation rate leads to a higher INP mass...)?*

The INP mass shown in Fig. 8 includes all particles (as always in this manuscript), not just those that will freeze, so it is the limiting factor. In other words, INP mass is independent of nucleation rate. We have clarified our terminology in the manuscript.

*- Fig. 9: Here the labels etc. in the figure and/or caption could be a bit more detailed. It is difficult to understand the plot like this without the text.*

We have clarified the figure (line styles, labels and caption) and the manuscript text (see below) so that there are now two sets of sensitivity tests (microphysics and de-coupled boundary layer).

"Here we examine the effects of microphysical (aerosol and cloud droplet sedimentation, and aerosol freezing immersion freezing of unactivated aerosol) and meteorological (de-coupled marine boundary layer) model considerations mentioned above. Figure 9 shows four test simulations and the high background marine INP simulation as the reference casefor the sedimentation and aerosol freezing simulationsthe test simulations divided into two groups according to the reference case. The reference case for the microphysical tests (enabled aerosol and cloud droplet sedimentation or aerosol freezing) is the simulation with high background INP concentration and marine INP emissions on. The effect of de-coupled boundary layermarine boundary layer (MBL) is tested by running simulations with and without based on the modified initial temperature and humidity profiles, which are explained below, when marine INP emissions are off or on (high background INP concentration for both simulations)."

*- p. 15, l. 293: "aerosol freezing" is unclear to me. Which freezing pathways are meant? Deposition and contact freezing?*

We clarify here (see below) and elsewhere in the manuscript that the aqueous aerosol freezes via immersion freezing, just like cloud droplets in the default simulations.

"Figure 9 shows that allowing aerosol  immersion freezing (freezing in the default simulations is limited to cloud droplets) has a small impact on ice crystal number concentration."

*- p. 16, l. 338: You write here that your results are limited to your simulations, that is also limited by the chosen case study. This could be discussed a lot more in detail (here and maybe even in the introduction and methods section).*

Writing that we are limited to our simulations includes not only the model setup based on the chosen case study but also other modelling assumptions and uncertainties (especially the ice nucleation scheme). Unfortunately, the other uncertainties dominate. We clarify that the results are limited to **the conditions used** in our simulations. Although we not focusing on ISDAC, we have extended the description of the case study in the methods-section. The case study represents a typical marine mixed-phase cloud, which could be seen elsewhere.

***Technical corrections:***

*- Values and units sometimes break over two lines, make sure to have a protected space between values and units to prevent this.*

This is now fixed.

*- p. 13, l. 275: I would suggest "cloud regions" instead of "clouds".*

It is now "cloud regions".

*- p. 13, l. 278: "Following..." is without a clear reference, the sentence like that does not seem complete.*

Removed "Following".

*- p. 15, l. 321: Change to "maintain the simulated mixed-phase cloud".*

Done.

**References**

Ahola, J., Korhonen, H., Tonttila, J., Romakkaniemi, S., Kokkola, H., and Raatikainen, T.: Modelling mixed-phase clouds with the large-eddy model UCLALES-SALSA, Atmos. Chem. Phys., 20, 11 639-11 654, https://doi.org/10.5194/acp-20-11639-2020, 2020.

Bullard, J. E., Baddock, M., Bradwell, T., Crusius, J., Darlington, E., Gaiero, D., Gassó, S., Gisladottir, G., Hodgkins, R., McCulloch, R., McKenna-Neuman, C., Mockford, T., Stewart, H., and Thorsteinsson, T: High-latitude dust in the Earth system, Rev. Geophys., 54, 520 447-485, https://doi.org/10.1002/2016RG000518, 2016.

Hartmann, M., Gong, X., Kecorius, S., van Pinxteren, M., Vogl, T., Welti, A., Wex, H., Zeppenfeld, S., Herrmann, H., Wiedensohler, A., and Stratmann, F.: Terrestrial or marine – indications towards the origin of ice-nucleating particles during melt season in the European Arctic up to 83.7 N, Atmos. Chem. Phys., 21, 11613-11636, https://doi.org/10.5194/acp-21-11613-2021, 2021.

McCluskey, C. S., DeMott, P. J., Ma, P.-L., and Burrows, S. M.: Numerical Representations of Marine Ice-Nucleating Particles in Remote Marine Environments Evaluated Against Observations, Geophys. Res. Lett., 46, 7838-7847, https://doi.org/https://doi.org/10.1029/2018GL081861, 2019.

Sanchez-Marroquin, A., Arnalds, O., Baustian-Dorsi, K. J., Browse, J., Dagsson-Waldhauserova, P., Harrison, A. D., Maters, E. C., Pringle, K. J., Vergara-Temprado, J., Burke, I. T., McQuaid, J. B., Carslaw, K. S., and Murray, B. J.: Iceland is an episodic source of atmospheric ice-nucleating particles relevant for mixed-phase clouds, Sci. Adv., 6, eaba8137, https://doi.org/10.1126/sciadv.aba8137, 2020.

Tobo, Y., Adachi, K, DeMott, P. J., Hill, T. C. J., Hamilton, D. S., 660 Mahowald, N. M., Nagatsuka, N., Ohata, S., Uetake, J., Kondo, Y., and Koike, M.: Glacially sourced dust as a potentially significant source of ice nucleating particles, Nat. Geosci., 12, 253-258, https://doi.org/10.1038/s41561-019-0314-x, 2019.

Wex, H., Huang, L., Zhang, W., Hung, H., Traversi, R., Becagli, S., Sheesley, R. J., Moffett, C. E., Barrett, T. E., Bossi, R., Skov, H., Hüerbein, A., Lubitz, J., Löffler, M., Linke, O., Hartmann, M., Herenz, P., and Stratmann, F.: Annual variability of ice-nucleating particle concentrations at different Arctic locations, Atmos. Chem. Phys., 19, 5293-5311, https://doi.org/10.5194/acp-19-5293-2019, 2019.

---

## Author Response (AR2)

We would like to thank Referee 1 for the useful comment, which helps us to improve the manuscript. Below is the original referee comment (*shown with italicized font*) and our reply is below that (standard font). Updates to the manuscript are also specified below (added and unchanged text).

*I would really like to thank the authors for the work they have done. The manuscript has substantially improved since the previous iteration, addressing most of the issues that the reviewers have raised. I recommend it for publication after a minor point is addressed.*

*Although the authors have added a comparison between the modelled and measured INPs in Sect. 2.2, I would like to see a more precise comparison (the data used for the comparison does not have enough information on temperature). For example, Creamean et al., 2018 measured INP concentrations in a close by location, covering the period of this study, reporting concentrations between 6\*10^-4 and 2\*10^-2 per L at a similar temperature to 258 K. I appreciate they used a filter, singular description-based technique which potentially would make all the existing "INP" to freeze. Would it possible to compare the modelled "INPs that produced ice" over a representative timescale to compare to those measurements by Creamean et al., 2018 or other relevant measurements?*

*Creamean, J. M., Kirpes, R. M., Pratt, K. A., Spada, N. J., Maahn, M., de Boer, G., Schnell, R. C., and China, S.: Marine and terrestrial influences on ice nucleating particles during continuous springtime measurements in an Arctic oilfield location, Atmos. Chem. Phys., 18, 18023–18042, https://doi.org/10.5194/acp-18-18023-2018, 2018*

We interpreted this comment so that it includes two parts: 1) information on temperature dependency of the ice nucleation scheme and 2) compare the modelled INP concentrations (simulated freezing based on instrument setting) to those measurements by Creamean et al. (2018) and others.

In the manuscript we focus on the 258 K (about −15 °C) temperature, because this is the minimum (cloud top) temperature seen in all our simulations. Moreover, the maximum in-cloud temperature is about 260 K, which means that ice nucleation takes place in a narrow temperature range. For this reason, we use the constant contact angle ice nucleation approach. The scheme is tuned for our case, so that it is valid close to −15 °C cloud temperatures. Extrapolation to other temperatures such as −20 °C or −10 °C often reported in the literature is not possible with this approach. For example, the CFDC instrument calculations (Sect. 2.2) showed that the ice crystal concentration resulting in from the maximum background aerosol would be 1.8 $L^{-1}$ at −15 °C, but an extrapolation to -10 °C would give zero ice crystals while all droplets (16.5 $L^{-1}$) would freeze at −20 °C. Because our parametrization is not valid for extrapolation and it is adjusted for the simulated cloud conditions focusing mostly on resulting reasonable cloud ice crystal number concentration, we are not examining the temperature dependency in the manuscript. We clarify this in Sect. 2.2 (line 184) with this addition:

…angle was increased from 0.50 to 0.57 to enhance freezing at these relatively high temperatures (see Appendix A in Ahola et al. (2020)). This value is within the range of 0.36–0.73 representing surface soil, quartz and sand (Khvorostyanov and Curry, 2000). Assuming a constant contact angle means that the ice nucleation parametrization is valid for a narrow temperature range which in our case means in-cloud temperatures of about 258K. For this reason, we are not examining temperature dependency, but focus on the 258K temperature.

Creamean et al. (2018) used a drop freezing cold plate technique to measure INP concentrations as a function of temperature. The cold plate was cooled variably within a 1–10 °C min$^{-1}$ range from room temperature until around −30 °C. As explained above, our ice nucleation approach is valid for about −15°C temperature, so we can simulate cooling up to that temperature. Following the CFDC instrument method described in the manuscript (Sect 2.2), but now integrating also over temperature down to −15°C gives ice crystal number concentrations of 2.0 and 0.5 L$^{-1}$ for the 1 and 10 °C min$^{-1}$ cooling rates, respectively. As it is expected, these numbers are consistent for with the 1.8 L$^{-1}$ obtained for the CFDC. Because the results are consistent and this goes quite far from our topic, there is no need to add these calculations to the manuscript.

What comes to the INP concentration values reported by Creamean et al. (2018), these agree with the previous findings as summarized by Murray et al. (Opinion: Cloud-phase climate feedback and the importance of ice-nucleating particles, Atmos. Chem. Phys., 21, 665-679, 2021). In fact, Creamean et al. (2018) is one of their sources when reviewing previous INP measurements. The values reported by Creamean et al. (2018) and more broadly Murray et al. (2021) are covered in our simulations, where the background INP concentration ranges from zero up to above-mentioned maximum of 1.8 L$^{-1}$ (as would be seen with the CFDC instrument).